# Prophylactic Salpingectomy during Hysterectomy for Benign Disease: A Prospective Study to Evaluate High-Grade Serous Ovarian Carcinoma Precursors

**DOI:** 10.3390/jcm12010296

**Published:** 2022-12-30

**Authors:** Garri Tchartchian, Bernd Bojahr, Lucas Heils, Harald Krentel, Rudy L. De Wilde

**Affiliations:** 1Clinic for Minimal Invasive Surgery, 14129 Berlin-Zehlendorf, Germany; 2Clinic for Gynecology and Breast Center, Universitätsklinikum Greifswald, 17489 Greifswald, Germany; 3Clinic of Gynecology, Obstetrics, Gynecological Oncology and Senology, Academic Teaching Hospital, Bethesda Hospital, 47053 Duisburg, Germany; 4Carl von Ossietzky University of Oldenburg, University Clinic for Gynecology, Pius-Hospital, 26121 Oldenburg, Germany

**Keywords:** opportunistic salpingectomy, hysterectomy, LASH, laparoscopic, p53, STIL

## Abstract

Recent findings suggest that high-grade serous ovarian cancer can originate in the fallopian tube. Not only has that made the identification of precursor lesions pivotal in early detection and prevention of these cancers, prophylactic salpingectomy alongside hysterectomy for benign indications has been increasingly proposed as well. The present prospective single-center study included 273 women who underwent opportunistic salpingectomy alongside laparoscopic supracervical hysterectomy. Uterine and tubal histopathological results as well as intra- and postoperative complications were evaluated. The complication rate was 3.3%, of which none were caused by salpingectomy. Uterine histopathology diagnosed 181 patients (66.8%) with uterine myomas, 60 patients (22.1%) with adenomyosis, 29 patients (10.7%) with adenomyomatosis, and, 1 patient (0.4%) without pathological abnormality. p53 signatures were detected in 221 right fallopian tubes (80.9%) and in 229 left tubes (83.9%). In total, 8 patients showed bilateral STIL (2.9%), whereas in 1 patient (0.4%) STIL was detected in the left tube only. No STIC were detected. Laparoscopic opportunistic salpingectomy is demonstrated to be both safe and feasible. It appears to be promising to reduce the risk for ovarian cancer, yet more studies are needed to undoubtedly confirm this.

## 1. Introduction

Ovarian cancer is the gynecological cancer with the highest mortality. One of the reasons is a lack of screening tools for early detection, which renders the majority of these cancers to be detected at an advanced stage only, after they have already spread to peritoneal surfaces [1]. Recent findings, however, suggest that high grade ovarian carcinogenesis can originate in the fallopian tube [2]. It is postulated that early precancerous lesions, called Secretory Cell Outgrowths (SCOUT), which consist of a string of at least 30 pseudostratified secretory epithelial cells without p53 mutation, develop into so-called p53 signatures, a sequence of at least 12 secretory cells with dense nuclear p53 expression. A more advanced type of early lesion is called Serous Tubal Intermediate Lesion (STIL); a spectrum of epithelial changes ranging from normal appearing tubal epithelium expressing p53 signatures and low proliferation index shown via Ki67, to lesions with increasing degrees of cytologic atypia that fall short of Serous Tubal Intraepithelial Carcinoma (STIC) [3,4]. STIC can advance from STILs and are defined as non-ciliated cells with the following morphological characteristics: nuclear hyperchromasia and atypia, mitotic figures, and nuclear stratification. Immunohistochemically, they exhibit increased staining for p53 and Ki67 [5,6].

Increasing evidence has been pointing at the potential metastasis of STIC in the ovaries and peritoneum, supporting the notion that ovarian carcinogenesis can have a tubal origin [7,8].

Altogether, interest to identify precursor lesions defined by both morphological and molecular characteristics has been growing. They could not only be used for early detection but also for the prevention of ovarian carcinoma. Furthermore, opportunistic prophylactic salpingectomy in benign hysterectomy while conserving the ovaries to avoid provoking menopause has been recommended by several leading institutions [9,10,11].

In this prospective single-center study, we evaluated tubal histopathological abnormalities (p53 signatures, Ki67 and STIC) as well as intra- and postoperative complications related to opportunistic salpingectomy (OS) performed during hysterectomy for benign uterine diseases.

## 2. Materials and Methods

### 2.1. Study Design and Patient Selection

Tubal histopathological results as well as intra- and postoperative complications of OS performed alongside LASH for benign uterine diseases were evaluated between 25 April 2017 and 23 March 2019 in a prospective single-center study. All surgeries were performed at the Clinic for Minimal Invasive Surgery (MIC-Klinik) in Berlin (Germany) by one of the expert in-house surgeons. During the study period, all patients with indication for hysterectomy were asked for informed consent to receive salpingectomy alongside LASH prior to surgery.

Indications for LASH were uterine fibroids associated with discomforting symptoms or with tendency towards growth or dysfunctional bleeding resistant to therapy (menorrhagia, metrorrhagia, dysmenorrhea, or hypermenorrhea). Patients who previously received (unilateral) salpingectomy were excluded from the study. Patients who received surgical treatment for malignancy (cervical, endometrial, or, ovarian tumors) were not included in the study. In order to exclude malignant abnormalities, a colposcopic and cytological evaluation of the cervix was conducted, for the purpose of which the Pap smear was not older than 12 months. In case of dysfunctional bleeding resistant to therapy or sonographic abnormalities with regards to the endometrium, a diagnostic hysteroscopy with fractionated abrasion was performed to exclude malignancy.

In total, 277 women complied with the inclusion criteria within our study period and were included in this prospective study. Four of these did not give consent to include their data which is why the present study evaluated the data of 273 patients.

### 2.2. Ethical Approval

The present prospective study received ethical approval from the ethical committee of the Medicine and Health Sciences department of the University Oldenburg (19 December 2016), as well as from the ethical committee of the Berlin Medical Board (21 February 2017).

### 2.3. Surgical Procedure

At the beginning of surgery, the adnexa were detached from the uterus so that the fallopian tubes remained intact throughout the operation. LASH was performed according to the customary manner [12]. As of 2012, LASH has been combined with the in-house established change-over technique [13,14]. This method ensures a better view and access to the uterus, which allows for uteri of almost any size to be safely removed [15]. Towards the end of LASH surgery, after detachment of the corpus uteri and peritonealisation of the cervix uteri, salpingectomy was performed. The tubes were not morcellated, yet were the first tissue to be removed from the abdominal cavity through a 10 mm trocar. This was followed by the morcellation of the corpus uteri. The left tube, right tube, and morcellated corpus uteri were separately labeled and forwarded to the histopathological institute for further analysis.

Routinely, when no complications occur, patients are discharged from the MIC Clinic 48 h post LASH surgery after which they are followed up via outpatient care.

### 2.4. Data Collection

This prospective study evaluated the medical files of our patient collective of 273 women. The age of patients, body mass index (BMI), parity, previous gynecological operations, and indication for surgery were recorded. We also documented the number and type of intra- and postoperative complications. Additionally, uterine and tubal histological data were collected. The SEE-FIM protocol by Medeiros et al. (2007) [16] was used to section the fimbria and consequently stain for p53 and Ki67. For the differential diagnosis between low and high grade precancerous lesions, we used the Ki67 index as described by Gurda et al. (2014): Ki67 < 10% and Ki67 > 10% are considered low and high grade, respectively [17].

### 2.5. Statistics

Demographic data were analyzed using Excel (Microsoft Corporation, 2018, Version 16.16.27) to calculate mean and standard deviation. The primary focus of this study was to assess tubal abnormalities (p53 and Ki67 signatures as well as STIC). As a secondary goal, we evaluated intra- and postoperative complications related to the performance of OS alongside LASH for benign indications. For descriptive results, we calculated percentage and number of patients within the total study population.

## 3. Results

### 3.1. Demographics and Preoperative Characteristics

Between 25 April 2017 and 23 March 2019, 273 women were treated with prophylactic salpingectomy alongside LASH for benign uterine diseases. Patients’ demographic data can be found in Table 1.

The mean age was 43.9 ± 4.2 years and mean BMI was 25.6 ± 5.2 kg/m^2^. Average parity was 1.3 ± 1.07. A total of 242 patients (88.6%) previously underwent surgery, and 91 patients (33.3%) had previous abdominal surgeries, whereas 166 (60.8%) received prior gynecological surgeries, which involved laparotomy in 63 cases. Indications for LASH surgery were uterus myomatosus in 194 patients (71.1%), bleeding disorders in 70 (25.6%), and endometriosis in 7 patients (2.6%).

### 3.2. Histopathology of Uteri and Fallopian Tubes

Uterine histopathological analysis was always benign (Table 2). Uterine samples of two patients were missing (*n* = 271). The majority of patients (181, 66.8%) were diagnosed with uterine myomas. Furthermore, histopathological results showed adenomyosis in 60 patients (22.1%), adenomyomatosis in 29 cases (10.7%), and no pathological abnormality in 1 patient (0.4%).

Table 3 shows the histopathological results of the fallopian tubes of our study collective of 273 patients. Primary findings included low-grade dysplasia in 115 (42.1%) right tubes and 118 (42.1%) left tubes, tubal hydatid cysts in 56 (20.5%) right tubes and 51 (18.7%) left tubes, tubal endometriosis in 1 (0.4%) right tube and in 3 (1.1%) left tubes, benign cyst adenoma in 2 (0.73%) right tubes, and no pathological abnormality in 1 (0.4%) right tube.

Staining for pre-cancerous lesion markers revealed p53 signatures in 221 right fallopian tubes (80.9%) and in 229 left tubes (83.9%). STIL was confirmed when the p53 signature coincided with a Ki67 staining higher than 10%. As such, eight patients showed bilateral STIL (2.9%), whereas in one patient (0.4%) STIL was detected in the left tube only. No STIC were detected.

### 3.3. Complications

Complications occurred in nine patients (complication rate of 3.3%) (Table 4). Intra-operative excessive bleeding with a bleeding volume of 700 mL occurred in two patients. Neither needed blood transfusion or further therapeutic intervention. Seven patients showed post-operative complications. In five of these cases, post-operative bleeding required therapeutic or surgical intervention; one patient had strong bleeding in the right parametrium and received a blood transfusion, three patients needed cooling with ice at the laparoscopic entry point, and one patient required re-laparoscopy because of decreased blood pressure and circulation, which was without further consequence after the bleeding at the left uterine artery was stopped (blood loss was 800 mL). None of the complications were caused by salpingectomy.

## 4. Discussion

In recent years, research has identified the distant fallopian tube as a potential origin site for serous cancers in women [18,19,20]. Interestingly, this appears to be true not only for women who show higher risk for ovarian and other pelvic cancers (e.g., BRCA1/2 mutations), but also for women without any positive predictive factors. Although the behavior of STICs may vary between women with and without BRCA mutations, they are otherwise identical in appearance and location [21]. Because ovarian cancers have a poor prognosis and screening tools for early detection have been lacking, it was suggested to make opportunistic bilateral salpingectomy a standard practice.

To preserve ovarian function until the age of natural menopause, several studies proposed early salpingectomy with delayed oophorectomy in patients with higher risk for ovarian cancers. As no significant increase in ovarian cancer incidence was observed, this approach appears to be a reasonable alternative [7,22].

Even though numerous studies have concluded that OS could effectively reduce the risk for ovarian cancer, it is still disputed [23,24]. Falconer et al. (2015) have recommended removal of fallopian tubes only for the general post-menopausal population [25]. Our results show that pre-cancerous p53 signatures were found in 80.9% of the right and in 83.9% of the left tubes. This is considerably higher than the prevalence found in the literature, where p53 signatures are detected between one fourth and half of tubal samples of patients with or without risk factors for ovarian cancer [1,4,26,27]. We found STILs in nine patients (in eight patients bilateral, in one unilateral) which is in line with the STIL prevalence found in other studies. None of the tubes of our patients showed STICs. This could be explained by the non-menopausal age of our patients (mean age was 43.9 ± 4.2 years) on the one hand and by the relatively small number of patients in our study on the other hand.

In the last decade, opportunistic salpingectomy has been increasingly performed and recommended, and is generally considered to be safe [28,29,30,31,32]. The laparoscopic approach in particular seems to cause no additional morbidity when OS is performed alongside hysterectomy [33,34]. This is in line with what we observed in the present study. In contrast, Collins et al. (2018) [35] observed a higher risk for minor complications, such as a slightly higher bleeding volume or longer hospital stay, when OS was combined with hysterectomy, although the correlation with salpingectomy was not significant. The vaginal approach appears to be more difficult with a success rate between 73.9 and 88% for experienced teams [36].

It is worth mentioning that the MIC Clinic in Berlin is a reference center for laparoscopic hysterectomies and myomectomies. Because of our high experience with LASH [37], our complication rate is very low [12]. As such, our study published in 2015, which evaluated 10,731 patients treated with LASH, revealed a complication rate of only 0.23 % [38]. Our four in-house surgeons each perform between 500 and 1000 LASH per year, thus our growing experience reduces the complication rate even more. As a reference, LASH complication rates in the literature range from 1% up to over 14% [39,40,41,42,43].

Because a delay of 1 year to the onset of menopause already results in a decreased risk of ischemic heart disease (hazard ratio, 0.98 per year; 95% confidence interval, 0.96–0.99) [44], it is imperative to discuss the uncertainty that still exists about the potential effect of OS on the time menopause starts. As such, Morelli et al. (2013) observed no significant difference in menopause markers (such as antimüllerian hormone AMH, Follicle Stimulating Hormone FSH, Antral Follicle Count AFC) [28]. Accordingly, Hanley et al. (2020) observed no difference as to when hormone replacement therapy was started when comparing women who underwent OS alongside hysterectomy vs hysterectomy only [45]. A recent meta-analysis concluded that salpingectomy does not appear to affect ovarian function, yet it may impair the ovarian reserve in the long run [46]. On the other hand, Collins et al. (2018) observed an increased risk for menopausal symptoms 1 year after surgery when bilateral salpingectomy was performed [35]. Piek et al. (2020) detected slightly lower AMH levels (marker for ovarian reserve) 6 months after hysterectomy with OS vs without [47]. These differences were not significant and the suitability of AMH only to predict the onset of menopause is questionable. Larger and long-term studies are needed to evaluate the impact of OS on the age of menopause, including the evaluation of other predictors or markers for menopause.

This study, as with all similar studies, is limited by the fact that OS requires consent from the patient, which can partially divert study populations from being representative of the entire target group. Obviously, all women need to receive all information before consenting to OS. This requires an in-depth discussion with patients so they fully grasp the risk-benefits based on the current knowledge, the effect it may have on menopause onset, as well as on psychosocial effects. Furthermore, because the current study only evaluates a relatively low number of patients (273) and the complication rate is low (9 patients, 3.3%), the evaluation of the correlation between preoperative and demographic characteristics with number and type of complications is not informative.

## 5. Conclusions

Laparoscopic opportunistic salpingectomy has been demonstrated to be both safe and feasible. It appears to be promising to reduce the risk for ovarian cancer, yet more studies are needed to undoubtedly confirm this. Finally, it is imperative to have an in-depth discussion with patients prior surgery on currently known risk-benefits, and the potential effect on menopausal onset and on psycho-social factors.

## Figures and Tables

**Table 1 jcm-12-00296-t001:** Preoperative Characteristics of the Study Population.

	Study Group (*n* = 273)
Age (years) mean ± SD	43.9 ± 4.2
Parity (*n*, %)	
Nullipara	73 (26.7)
Unipara	78 (28.6)
Bipara	89 (32.6)
Tripara	26 (9.5)
>Tripara	7 (2.6)
sectio caesarea	50 (18.3)
mean ± SD (range)	1.3 ± 1.07
Body mass index (kg/m^2^) mean ± SD	25.6 ± 5.2
Previous Surgery (*n*, %)	242 (88.6)
Previous abdominal surgery (*n*, %)	91 (33.3)
Previous gynecological surgery (*n*, %)	166 (60.8)
with Laparotomy	63 (23.1)
Indication for surgery (*n*, %)	*n* = 273
uterus myomatosus	194 (71.1)
bleeding disorders	70 (25.6)
endometriosis	7 (2.6)

**Table 2 jcm-12-00296-t002:** Main Histopathological Uterine Results.

	*n* = 271
Uterus myomas (*n*, %)	181 (66.8)
Adenomyosis (*n*, %)	60 (22.1)
Adenomyomatosis (*n*, %)	29 (10.7)
No pathological abnormality (*n*,%)	1 (0.4)

**Table 3 jcm-12-00296-t003:** Histopathological Results in Fallopian Tubes.

*n* = 273	Right Tube	Left Tube
Primary findings (*n*, %)		
Low-grade dysplasia	115 (42.1)	118 (42.1)
Tubal hydatid cyst	56 (20.5)	51 (18.7)
Tubal endometriosis	1 (0.4)	3 (1.1)
Benign cystadenoma	2 (0.73)	0 (0.0)
No pathological abnormality	1 (0.4)	0 (0.0)
Pre-cancerous tubal lesion (*n*, %)		
p53 signature	221 (80.9)	229 (83.9)
STIL (p53 signature + Ki67 > 10%)	8 (2.9)	9 (3.3)
STIC	0 (0.0)	0 (0.0)

**Table 4 jcm-12-00296-t004:** Complications of LASH with OS in our Study Population.

Complications	*n* = 9 (3.3%)
Intra-operative	
excessive bleeding	2
Post-operative	
strong bleeding in right parametrium	1
bleeding at laparoscopic entry point	3
bleeding at left uterine artery	1

## Data Availability

The data presented in this study are available on request from the corresponding author. The data are not publicly available due to privacy restrictions.

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
