# Peer review of "Prophylactic Salpingectomy during Hysterectomy for Benign Disease: A Prospective Study to Evaluate High-Grade Serous Ovarian Carcinoma Precursors"

_jcm, 2022, doi:10.3390/jcm12010296_

Round 1

Reviewer 1 Report

The authors of the current manuscript discuss one of the most important topics in ovarian cancer(OC) therapy and surgery.  Opportunistic salpingo-oophorectomy (OS) is assigned for patients with a hereditary predisposition to OC development and reduces the risk of developing ovarian and fallopian tube cancer by up to 80%. 

However up to 80% of high-grade serous carcinomas occur in patients without germline pathogenic nucleotide variants, so they need a different strategy to prevent OC.

The authors suggest that OS might be an appropriate strategy for such patients. 

I have several concerns regarding the results presented in the current manuscript:

1. It would be more informative if the authors included Ki67 indexes in table 3, as well as the number of STIC cases

2. The preoperative characteristics of the study population (table 1) might be more extended, e.g., including such characteristics as menopausal status and family history. Consider analyzing the connection between these characteristics and the distribution of the complications with Fisher's exact test. 

3. It would be more informative if the authors would add the observation of the postsurgical state of the patients, e.g. days of hospital stay, bleeding volume  

Author Response

We appreciate your remarks and suggestions. Below, we include our responses to your comments as well as how we adapted and elaborated on our manuscript. Furthermore, our manuscript underwent extensive English revision.

1. It would be more informative if the authors included Ki67 indexes in table 3, as well as the number of STIC cases

--> We have included both in Table 3.

2. The preoperative characteristics of the study population (table 1) might be more extended, e.g., including such characteristics as menopausal status and family history. Consider analyzing the connection between these characteristics and the distribution of the complications with Fisher's exact test.

 --> The demographic and preoperative characteristics we chose to collect are those who are standardly chosen in Germany. However, we are planning a next study with a larger number of patients to further evaluate this topic and will plan to collect these and other data as well.

 --> In our next study with a larger number of patients, we agree that the evaluation of the connection between these characteristics and the complications will be very informative. The current study however only evaluates a relatively low number of patients (273) and the complication rate is very low (9 patients, 3.3 %).

We’d like to add that all surgeries were performed at the MIC Clinic in Berlin and the MIC clinic is a reference center for laparoscopic hysterectomies & myomectomies. Because of our high experience with LASH, our complication rate is very low: our study published in 2015 which evaluated 10731 patients treated with LASH revealed a complication rate of only 0.23 %. By now, we have performed over 19.000 LASH at our clinic and our growing experience reduces our complications rate even more.

3. It would be more informative if the authors would add the observation of the postsurgical state of the patients, e.g. days of hospital stay, bleeding volume

--> Routinely, when no complications occur, patients are discharged from the MIC Clinic 48 hours post-surgery after which they are followed up via outpatient care. We have added this information to our manuscript.

We plan to collect duration of hospital stay for patients with complications in our next study.

--> When loss of blood does not exceed 100 mL, the bleeding volume is not recorded at our clinic. Only for the few cases where complications included blood loss, we recorded the blood loss volume (700 mL for 2 patients intra-operative and 800 mL for 1 patient post-operative).

Reviewer 2 Report

the paper reports the results of the pathological analysis of the fallopian tubes of 273 women who underwent an opportunistic salpingectomy  during  laparoscopic supracervical hysterectomy.A p53 signature was detected in more than 220 tubes,STIL in 9 patients while  STIC  was not detected .No complications related  to the salpingectomy  were reported.The clinical value of those results is uncertain because the prognostic significance  of a p53 signature  if the fimbria is unknown

Author Response

We appreciate your remarks and suggestions. Below, we include our responses to your comment as well as how we adapted and elaborated on our manuscript. Furthermore, our manuscript underwent extensive English revision.

We became aware of this fact at the end of the study. We think that STICs were not present in our study population because of the non-menopausal age of our patients on the one hand and because of the relatively small number of patients on the other hand.

We suspect that with a number of patients between 800 and 1000, our results would be different. For these and other reasons, we are planning a next study with a larger number of patients to further evaluate this topic.

We have added this potential explanation of zero detected STICs to our manuscript.

Reviewer 3 Report

It is well designed prospective clinical study and the authors could show the significance of tubectomy for prevention of malignant tumor development from the serous tubal intermediate lesion (STIL) of the residual Fallopian tubes. It was approved by demonstration of changes in removed Fallopian tubes in this study. There were 8 (2.9) and 9 (3.3) cases of STIL among right and left Fallopian tubes with subsequent increased p53 signature 221 (80.9) 229 (83.9) respectively. Findings of this study can be useful for clinicians to choose treatment modalities for gynecological patients. The study well designed and manuscript was prepared sufficiently for publication.  

Author Response

We are grateful for the appreciation of our research.

We'd like to mention that our manuscript underwent extensive English revision.

Round 2

Reviewer 1 Report

"In our next study with a larger number of patients, we agree that the evaluation of the connection between these characteristics and the complications will be very informative. The current study however only evaluates a relatively low number of patients (273) and the complication rate is very low (9 patients, 3.3 %)."

Please, add this information to the discussion part, where you describe the limitations of the study

We’d like to add that all surgeries were performed at the MIC Clinic in Berlin and the MIC clinic is a reference center for laparoscopic hysterectomies & myomectomies. Because of our high experience with LASH, our complication rate is very low: our study published in 2015 which evaluated 10731 patients treated with LASH revealed a complication rate of only 0.23 %. By now, we have performed over 19.000 LASH at our clinic and our growing experience reduces our complications rate even more.

To evidence that the complication rate in your clinic is very low, please, provide a comparison of your frequency with data from other studies with similar directions.  Please, provide the range of complication rates for your group of patients (non-menopausal women)

Author Response

“In our next study with a larger number of patients, we agree that the evaluation of the connection between these characteristics and the complications will be very informative. The current study however only evaluates a relatively low number of patients (273) and the complication rate is very low (9 patients, 3.3 %).”

 Please, add this information to the discussion part, where you describe the limitations of the study

 --> We have added this information as suggested.

“We’d like to add that all surgeries were performed at the MIC Clinic in Berlin and the MIC clinic is a reference center for laparoscopic hysterectomies & myomectomies. Because of our high experience with LASH, our complication rate is very low: our study published in 2015 which evaluated 10731 patients treated with LASH revealed a complication rate of only 0.23 %. By now, we have performed over 19.000 LASH at our clinic and our growing experience reduces our complications rate even more.”

To evidence that the complication rate in your clinic is very low, please, provide a comparison of your frequency with data from other studies with similar directions.  Please, provide the range of complication rates for your group of patients (non-menopausal women)

 --> We have added this information in a paragraph in the Discussion.

Reviewer 2 Report

no additional comments 

Author Response

no additional comments

--> We appreciate your review and suggestions which have helped improve our manuscript.
